# Draft genome of the bluefin tuna blood fluke, *Cardicola forsteri*

**Lachlan Coff**[1¤], **Andrew J. Guy**[1,2], **Bronwyn E. Campbell**[1], **Barbara F. Nowak**[1,3], **Paul A. Ramsland**[1,4,5], **Nathan J. Bott**[1] *

1 School of Science, STEM College, RMIT University, Bundoora, Victoria, Australia, 2 ZiP Diagnostics, Collingwood, Victoria, Australia, 3 Institute for Marine and Antarctic Studies, University of Tasmania, Launceston, Tasmania, Australia, 4 Department of Immunology, Monash University, Melbourne, Victoria, Australia, 5 Department of Surgery, Austin Health, University of Melbourne, Heidelberg, Victoria, Australia

¤ Current address: Australian Centre for Disease Preparedness, CSIRO, East Geelong, Victoria, Australia
* Nathan.Bott@rmit.edu.au

**Data Availability Statement:** The raw sequence data and genome assembly generated in this study have been submitted to the NCBI BioProject database (https://www.ncbi.nlm.nih.gov/bioproject/) under the accession number PRJNA810749.

## Abstract

The blood fluke *Cardicola forsteri* (Trematoda: Aporocotylidae) is a pathogen of ranched bluefin tuna in Japan and Australia. Genomics of *Cardicola* spp. have thus far been limited to molecular phylogenetics of select gene sequences. In this study, sequencing of the *C. forsteri* genome was performed using Illumina short-read and Oxford Nanopore long-read technologies. The sequences were assembled *de novo* using a hybrid of short and long reads, which produced a high-quality contig-level assembly (N50 > 430 kb and L50 = 138). The assembly was also relatively complete and unfragmented, comprising 66% and 7.2% complete and fragmented metazoan Benchmarking Universal Single-Copy Orthologs (BUSCOs), respectively. A large portion (> 55%) of the genome was made up of intergenic repetitive elements, primarily long interspersed nuclear elements (LINEs), while protein-coding regions cover > 6%. Gene prediction identified 8,564 hypothetical polypeptides, > 77% of which are homologous to published sequences of other species. The identification of select putative proteins, including cathepsins, calpains, tetraspanins, and glycosyltransferases is discussed. This is the first genome assembly of any aporocotylid, a major step toward understanding of the biology of this family of fish blood flukes and their interactions within hosts.

## Introduction

*Cardicola forsteri* (Trematoda: Aporocotylidae) is a blood fluke that infects bluefin tunas (*Thunnus* spp.) [1, 2]. *C. forsteri* parasitises *Thunnus* spp. hearts in its adult life cycle stage, while adult *C. orientalis* are found in the branchial arteries of the gills [3, 4]. Additionally, the co-infecting species *C. opisthorchis* is found in the hearts of Pacific bluefin tuna (PBT, *Thunnus orientalis*) [5], but has not been found in southern bluefin tuna (SBT, *Thunnus maccoyii*). While less prevalent in wild populations, infections with *Cardicola* spp. represents the most significant health issue for ranched bluefin tuna, a multimillion-dollar industry operating in Japan, Australia, and the Mediterranean [6]. Treatment with praziquantel (PZQ) has significantly reduced mortalities of SBT since its implementation in 2013, and *C. forsteri* is now the

Other data are described in the Supplemental Material. All protein names used in the paper adhere to approved nomenclature guidelines.

**Funding:** This work was funded by the Fisheries Research and Development Corporaton (2018-170) to NJB. The funders had no role in study design, data collection and analysis, decision to publish, or preparation of the manuscript.

**Competing interests:** The authors have declared that no competing interests exist.

dominant *Cardicola* spp. infecting SBT [7]. Despite the success of PZQ in the treatment of *Cardicola* infections, the required holding period and the risk of resistance has prompted research into alternative control measures. A variety of animal husbandry practices can also reduce the burden of helminthiases in ranched bluefin tunas, but additional pharmacological and immunological interventions would benefit the industry [8–10]. However, very little is known about *Cardicola* spp. beyond morphological characterisations and identifications of their intermediate hosts [11–13]. A full genome sequence of *C. forsteri* could help to answer fundamental questions of its biology and facilitate transcriptomic and proteomic investigations.

The genomes of the human blood flukes *Schistosoma mansoni* and *S. japonicum* (Trematoda: Schistosomatidae), were first published in 2009 using chain-termination (Sanger) sequencing technology [14, 15]. The advent of Illumina sequencing dramatically reduced sequencing costs, so genomes for other digenean trematodes, including *S. haematobium*, *Fasciola hepatica*, and *F. gigantica*, were published in the 2010s [16–18]. However, despite the relatively high base call accuracy of Illumina sequencing, its short-read lengths alone cannot bridge the large repeat regions typical of these genomes, so these initial assemblies are relatively discontiguous and fragmented. The addition of Third-Generation sequencing technologies, namely PacBio single molecule real time sequencing (SMRT) and Oxford Nanopore sequencing, has greatly improved these assemblies. The *S. mansoni* genome is now in its ninth revision (GCA_000237925.5), > 95% of which is assembled into seven autosomes and two sex chromosomes [19].

*Cardicola* was first diagnosed by Robert B. Short in 1953 [20], and 34 novel species have since been added to the genus, primarily based on morphological characterisations, making *Cardicola* the most speciose genus within the Aporocotylidae. Until now, only ribosomal 28S, internal transcribed spacer 2 (ITS-2), and mitochondrial cytochrome c oxidase subunit I (*cox*1) sequences of *C. forsteri* have been published, and these were recently used to demonstrate polyphyly in the genus and to reclassify other species in the genus [21]. Whole genome sequencing (WGS) could further aid in molecular phylogenetics of Aporocotylidae and their evolutionary relationships to other trematodes. Additionally, putative functional information can be mined from hypothetical proteins, which would direct further studies into the structural biology and host–parasite interactions of *Cardicola* spp.

Vaccines have been implemented in aquaculture since the 1980s to control a variety of infections [22], and immunization of farmed bluefin tunas with immunodominant antigens could be a viable control measure for infections with *Cardicola* spp. [1]. While there are currently no approved vaccines against helminthiases for fishes, several targets are undergoing human clinical trials, and some vaccines are approved to control helminthiases in mammalian livestock [23]. These vaccines chiefly interfere with digestive enzymes or target the surface tegument, which is the primary interface of platyhelminths accessible by the host immune system. A functionally annotated genome assembly of *C. forsteri* would facilitate further research into host–parasite interactions and rational vaccine design by homology to known vaccine targets in other digeneans. In this study, we present the first draft of the *C. forsteri* genome, assembled *de novo* from a hybrid of short-read (Illumina) and long-read (Oxford Nanopore) sequences. Predicted genes were functionally annotated, and putative glycosyltransferases and vaccine targets are discussed.

## Methods

### Specimen collection

Whole adult *Cardicola forsteri* specimens were flushed from the hearts of wild-caught southern bluefin tuna (SBT, *T. maccoyii*), ranched in the lower Spencer Gulf, according to the protocol

described by Aiken et al. [24]. These SBT were from untreated pontoons collected during the 2019 harvest, as per Power et al. [25]. Sampling was performed during harvesting under animal ethics approval (RMIT Animal Ethics Committee #22802) from specimens after euthanasia, which was performed by commercial SBT companies using industry best-practice techniques. The *C. forsteri* specimens were stored in RNAlater® at -20 ˚C.

## Genomic DNA extraction and sequencing

RNAlater® was completely washed from the specimens with Tris-buffered saline (TBS, 20 mM Tris, 150 mM NaCl, pH 7.6). For Illumina short-read sequencing, genomic (g)DNA from two adult specimens was extracted separately using the ISOLATE II Genomic DNA Kit (Bioline), according to the manufacturer's instructions. The yield of double-stranded (ds)DNA in each sample was measured by Qubit™ 4 Fluorometer (Thermo Fisher Scientific) using the dsDNA HS Assay Kit, according to the manufacturer's instructions. The purity of each sample was indicated with A260/280 and A260/230 ratios using the NanoDrop™ One microvolume spectrophotometer (Thermo Fisher Scientific), according to the manufacturer's instructions. These two samples were sent to the Ramaciotti Centre for Genomics for library preparation and sequencing. The short-read libraries from each sample were prepared using Nextera DNA Flex Library Prep Kit (Illumina®, Inc.), according to the manufacturer's instructions. Each sample was sequenced in paired ends of 150-bp lengths using the NextSeq® 500 System, which yields 300–550 bp insert size (Illumina®, Inc.).

For Nanopore long-read sequencing, high-molecular weight (HMW) gDNA was extracted from approximately 30 pooled adult specimens using the MagAttract® HMW DNA Kit (QIAGEN), according to the manufacturer's instructions. The yield of dsDNA in the sample was measured by Qubit™ 4 Fluorometer, according to the manufacturer's instructions. Purity was indicated with $A_{260}/A_{280}$ and $A_{260}/A_{230}$ ratios using the NanoDrop™, according to the manufacturer's instructions. Fragmentation was assessed using the TapeStation automated electrophoresis system (Agilent Technologies, Inc.), according to the manufacturer's instructions. This sample was also sent to the Ramaciotti Centre for Genomics for library preparation and sequencing. The HMW gDNA extracted from the pooled specimens was sequenced with GridION MK1 (Oxford Nanopore Technologies Ltd.), using the FLO-MIN106 flow cell and Ligation Sequencing Kit (SQK-LSK110). Long-read sequences were base called using Guppy v4.3.4 High-Accuracy model [26].

## Trimming and removal of host contaminants from the short-read library

Base call quality scores were assessed using FastQC. To improve basecall accuracy, four nucleotides were trimmed from the 3′ end of each paired-end read in the short-read library using Trimmomatic v0.36 [27].

Each short-read sequence was queried against the genomes of Pacific bluefin tuna (*T. orientalis*, GCA_009176245.1) and *S. mansoni* (GCA_000237925.4) using pBLAT v2.5 with default parameters [28, 29]. These reference genomes were chosen as the closest relatives of the host (*T. maccoyii*) and parasite (*C. forsteri*), respectively, with high-quality published genomes [29]. Of the matching reads with high query-coverages (116–146 bp), those that matched only with the *T. orientalis* and not the *S. mansoni* genome were removed from the short-read library. Species identity was confirmed *in silico* using pBLAT, with complete sequence identity of *C. forsteri* ITS-2 (AB742428.1) and 28S (AB742426.1) nucleotide sequences to the short-read library [3, 21].

## Short-read assembly and *k*-mer optimisation

Short reads for each specimen were assembled using the Hamiltonian de Bruijn graph assembler ABySS v2.1.5 [30], with *k*-mers from 50 to 102 and a minimum *k*-mer depth of four (kc parameter). The contiguity statistics for each assembly were then examined, and the *k*-mer that produced the highest N50 and *E*-size, and lowest L50 for each of the specimens was selected as the optimal *k*-mer for short-read assembly of that specimen.

## Estimation of the size, ploidy, and heterozygosity of the *C. forsteri* genome

A *k*-mer depth plot for each specimen was constructed with Jellyfish v2.3.0 [31], and this was used to determine the peak *k*-mer depth ($D'$) for estimation of the haploid genome size, according to the following equations:

$$D = \frac{D'l}{l - k + 1}$$

and

$$G = \frac{n_{read}(l - k + 1)}{D'} = \frac{n_{k-mer}}{D'} = \frac{n_{base}}{D}$$

where $G$ is haploid genome size, $D$ is read depth, $l$ is average read length, and $k$ is *k*-mer length [32].

The *k*-mer depth plot was also used as the input for GenomeScope [33], which estimated the haploid size, repeat length, and heterozygosity of the *C. forsteri* genome. Ploidy was estimated from this *k*-mer depth plot using Smudgeplot v0.2.5 [34].

## *De novo* assembly of hybrid contigs

The genome was assembled *de novo* using the short-read contigs from each of the ABySS assemblies and long reads as inputs for Wengan v0.2 [35]. Briefly, Wengan corrects short-read contigs by splitting chimeric contigs that lack paired-end support. Then, a synthetic mate-pair library is constructed from the long reads, which are mapped to the corrected short-read contigs, spanning the gaps and repeat regions. This information, along with the full long reads, is used to construct a synthetic scaffolding graph and, ultimately, hybrid contigs. The Wengan input accounted for the size and coverage of the long reads: large genome size (> 500 Mb, -g parameter), insert sizes of synthetic mate-pair reads 0.5–20 kb (-i parameter), 3 long-reads required to keep a potentially erroneous mate-edge (-N parameter), and 5 kb as the minimum length of reduced paths to convert them to physical fragments (-P parameter).

## Assessment of contiguity and completeness of the genome assembly

Contiguity statistics of the hybrid contigs for each assembly were calculated using QUAST v5.0.2 [36]. The completeness of each assembly was assessed using BUSCO v5.2.2 [37] against the metazoa_odb10 database, which comprises 954 Benchmarking Universal Single-Copy Orthologs (BUSCOs). These statistics were compared with those of published genome assemblies of other trematodes, using the same metazoan database for a like-for-like comparison (S1 Table).

## Repeat masking and annotation of the draft genome

A *de novo* repeat library was compiled from the *C. forsteri* genome assembly using RepeatModeler v2.0.2a [38], which comprises RECON v1.08 [39], RepeatScout v1.0.6 [40], LTRharvest

v1.5.9 [41], and LTR_retriever v2.9.0 [42], and these repetitive elements were softmasked with RepeatMasker v4.1.2-p1 [43].

For consistency, the genome assembly with the greater contiguity and completeness was selected from the two for gene prediction and functional annotation. *Ab initio* and homology-based gene prediction were performed using the BRAKER2 pipeline [44]. This pipeline firstly executed the ProtHint pipeline, which generated a set of seed genes from the *C. forsteri* genome assembly using the self-training *ab initio* gene prediction tool GeneMark-ES [45]. These seed genes were then translated into seed proteins, which were queried against a database of reference proteins (*S. mansoni*, GCF_000237925.1) using DIAMOND [46]. The output from ProtHint was ultimately used to train GeneMark-EP+ [47], which generated a set of anchored genes to train AUGUSTUS v3.4.0 [48] for the final output of predicted genes. Hypothetical polypeptide sequences were translated from the predicted genes, and the completeness of the gene set was assessed using BUSCO in protein mode.

For functional annotation, the hypothetical polypeptide sequences were queried using BLASTp (*E*-value $\leq 10^{-5}$) against the curated Swiss-Prot database [49]. Select sequences were further characterised as putative proteases, glycosyltransferases, ribonucleases, calpains, cation channels, tetraspanins (TSPs), glutathione *S*-transferases (GSTs), TGF-β homologs, and fatty acid-binding proteins (FABPs) based on BLASTp matches to the National Center for Biotechnology Information's (NCBI's) non-redundant protein sequences (NR) database and motif identification using HMMER v3.2.2 [50] with the Pfam database v35.0 [51] and CAZy database [52]. Signal peptides were predicted using SignalP 6.0 (probability $\geq 0.95$) [53], transmembrane domains were predicted using DeepTMHMM v0.0.47 (https://biolib.com/DTU/DeepTMHMM), and N-glycosylation sites were predicted using NetNGlyc v1.0 [54].

### Phylogenetic analyses

Outgroup-rooted phylogenetic trees of *C. forsteri* cathepsins and fucosyltransferases were constructed using the maximum likelihood method, with homologous proteins from SBT set as the outgroup. The protein sequences were aligned with MUSCLE v3.8.1551 [55], and these alignments were constructed as phylogenetic trees using RAxML v8.2.12 [56] with the Gamma model of rate heterogeneity, the WAG amino acid substitution model, and the majority rule consensus tree criterion.

## Results

### Sequencing and assembly

Approximately 450 ng of dsDNA was extracted from each *C. forsteri* specimen selected for short-read sequencing, and Illumina sequencing yielded 86 million paired-end reads from Specimen 1 and 127 million from Specimen 2 (Table 1). Due to the higher read output and the contiguity of the resulting assemblies, only the Specimen 2 assembly is reported here, but more information on the Specimen 1 assembly is available in S1 Table. Approximately 0.2–0.3% of short reads were likely derived from the host (*T. maccoyii*) and were removed from the library prior to assembly. The optimal *k*-mer length was found to be 77, based on the N50, *E*-size, and L50 of the short-read assemblies (Fig 1A). Using this optimal *k*-mer, the peak *k*-mer depth (***D'***) is 15, followed by a secondary peak, which indicates a large number of repetitive bases (Fig 1B). The haploid genome size was estimated to be 530–600 Mb by calculating the number of *k*-mers over peak *k*-mer depth, whereas GenomeScope estimated 230–236 Mb, with 25–30 Mb (12–15%) of repetitive elements. Smudgeplot proposed *C. forsteri* to be diploid in its adult life cycle stage, based on the grouping of *k*-mer pairs into haplotypes. As the genome size and ploidy of *C. forsteri* have not been empirically validated, the GenomeScope

**Table 1. Statistics from short-read Illumina paired-end sequencing of *Cardicola forsteri*.**

| | |
|---|---|
| **Number of short reads** | 126,882,928 |
| **Number of short reads derived from the host** | 278,175 (0.22%) |
| **Optimal *k*-mer** | 77 |
| **Estimated haploid genome size (Mb)** | 230 |
| **Estimated short-read coverage** | 80× |
| **Estimated repeat length (Mb)** | 25 (12%) |
| **Estimated heterozygosity (%)** | 0.14 |
| **Assembled genome size (bp)** [a] | 216,531,548 (94%) |

[a] Total sequence length of the hybrid assembly.

estimate was selected for further calculations. Using the GenomeScope estimate of haploid genome size, short-read sequence coverage is 80×.

For long-read sequencing, approximately 19 µg of dsDNA was extracted from pooled *C. forsteri* specimens with a DNA integrity number (DIN) of 6.9, which indicates moderate degradation, and a modal fragment size of 15,640 bp (5–58 kb range). Nanopore sequencing yielded 4.97 million reads with an N50 of 5.5 kb and a total of 13.68 Gb (90.79% passed). Long-read coverage was estimated to be > 53×, using the haploid genome size estimate of 230–236 Mb. The short-read library was assembled with ABySS, and these short-read contigs were combined with the pooled long-read data and assembled with Wengan. The size of the final hybrid assembly is 217 Mb. The GC content of the *C. forsteri* genome is > 28%, which is lower than that of other platyhelminths (schistosome GC content > 34%) [57].

## Contiguity and completeness of hybrid assembly

An important quality metric of genome assemblies is contiguity, with larger contigs indicating a less fragmented assembly. The assembly is highly contiguous, with 1,532 hybrid (assembled from short and long reads) contigs (N50 = 430,422 and L50 = 138) (Fig 2A), and the largest contig is > 3 Mb. These contigs are more contiguous than those of the latest *F. hepatica* assembly (S1 Table), which comprises only short-read sequences. The BUSCO (Benchmarking Universal Single-Copy Ortholog) analysis indicates a relatively complete assembly, with 66.0% of metazoan BUSCOs found complete in the assembly and only 7.2% fragmented (Fig 2B). This

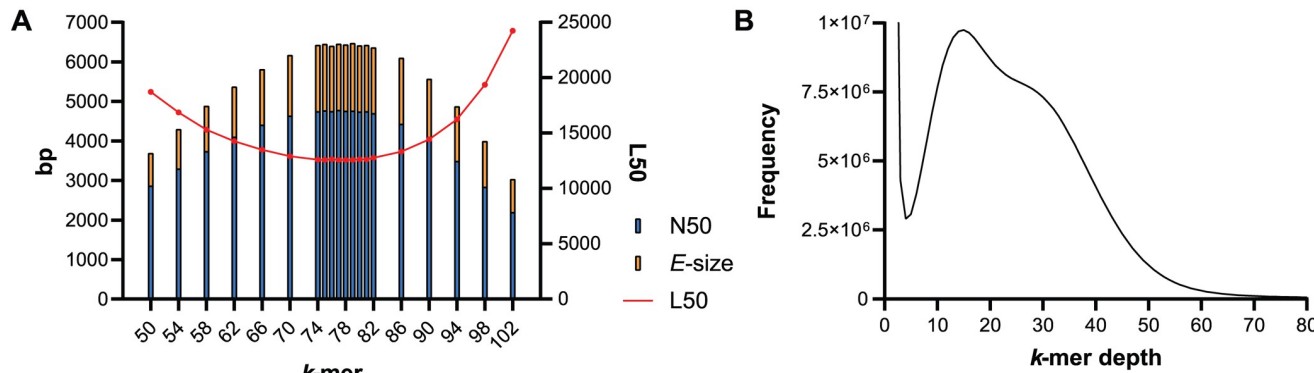

**Fig 1. Analyses of *k*-mers in the *Cardicola forsteri* short-read library.** (A) Contiguity statistics for the short-read assembly of different *k*-mers. N50 and *E*-size are shown as an overlaid column graph, while L50 is shown as an overlaid line graph with a secondary *y*-axis. (B) Plot of *k*-mer depth using the optimal *k*-mer 77.

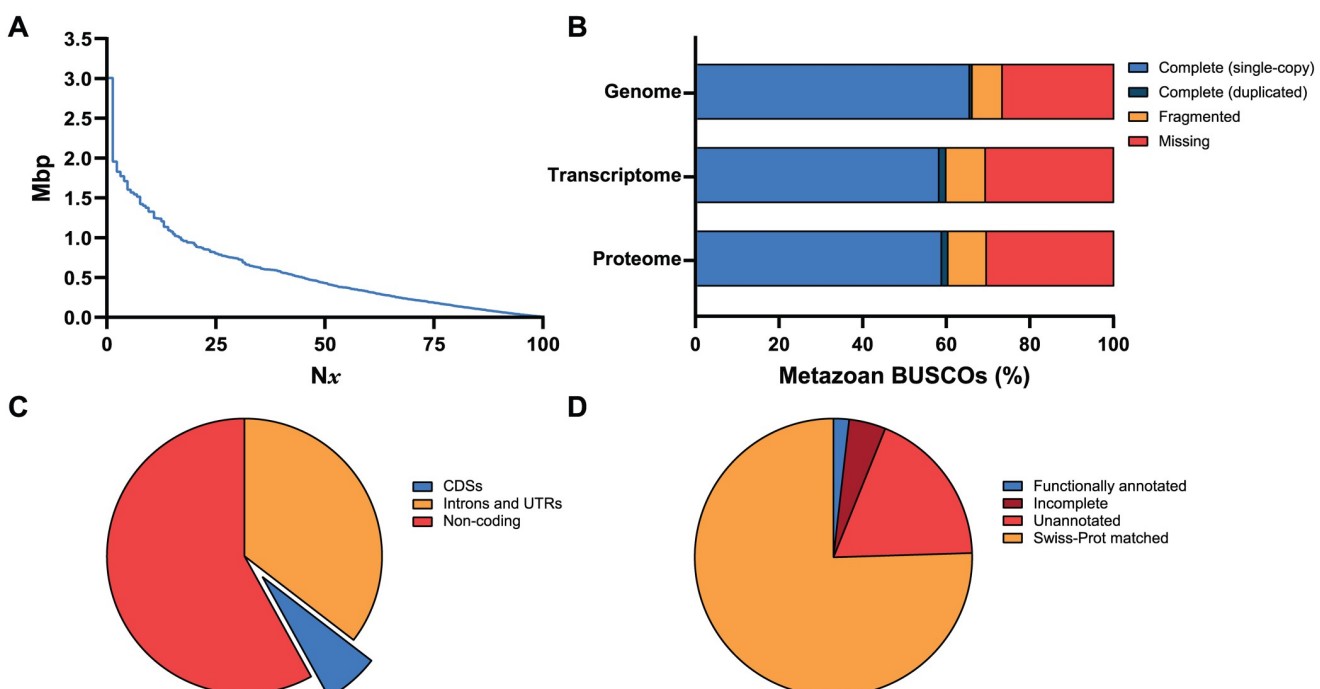

**Fig 2. Contiguity, completeness, and gene prediction and annotation statistics of the *Cardicola forsteri* genome assemblies.** (A) N*x* plot of the genome assembly, depicting the contig size distribution. (B) BUSCO statistics for the genome assembly, and the hypothetical transcripts and gene set (proteins), measured as a proportion of complete, fragmented, and missing BUSCOs from the metazoa_odb10 database. (C) Pie chart showing the sizes (bp) of predicted CDSs, introns, and UTRs as a proportion of the genome (*n* = 216,531,548). (D) Pie chart showing the number of hypothetical polypeptides (*n* = 8,564) that were functionally annotated, matched to the Swiss-Prot database or not (unannotated), and incomplete. BUSCOs, Benchmarking Universal Single-Copy Orthologs; CDS, coding sequence; UTR, untranslated region.

also supports *C. forsteri* as a more complete assembly than that of *F. hepatica*, which has 65.4% and 10.5% complete and fragmented BUSCOs, respectively. For further comparison, *S. mansoni* BUSCOs are 71.5% complete and 5.0% fragmented.

## Repetitive elements

Contrary to the estimates of GenomeScope, a large proportion of the *C. forsteri* genome was identified by RepeatModeler as repetitive and was masked prior to genome annotation. These repetitive elements comprise 124,057,400 bp (57.29%) of the assembly (Table 2). This

**Table 2. Repetitive elements classified by repeat modeler and masked prior to the annotation of the *Cardicola forsteri* genome.**

| Repetitive elements | Number | Length (Mb) |
|---|---|---|
| LINEs | 173,788 | 50.39 (23.27%) |
| LTRs | 10,709 | 3.82 (1.76%) |
| DNA transposons | 33,663 | 12.04 (5.56%) |
| Small RNA repeats | 30,817 | 2.85 (1.32%) |
| Low-complexity repeats | 137,547 | 8.29 (3.83%) |
| Unclassified repeats | 269,189 | 46.67 (21.55%) |
| Total | 655,713 | 124.06 (57.29%) |

LINEs, long interspersed nuclear elements; LTRs, long terminal repeats.

**Table 3. Gene prediction statistics of the *Cardicola forsteri* genome assembly.**

|  | Number | Total length (bp) | Average length (bp) |
|---|---|---|---|
| **Genes** | 8,564 | 90,783,647 (41.93%) | 10,600 |
| **CDSs** | 8,564 | 14,137,916 (6.53%) | 1,650 |
| **Exons** | 57,052 | 14,747,359 (6.81%) | 258 |
| **Introns** | 48,268 | 76,092,320 (35.14%) | 1,576 |
| **5′ UTRs** | 203 | 405,498 (0.19%) | 3 |
| **3′ UTRs** | 139 | 203,945 (0.09%) | 3 |

CDS, coding sequence; UTR, untranslated region.

proportion is higher than most other trematode genomes (40–54%), with the exception of *F. gigantica* (~ 70%) [57]. Long interspersed nuclear elements (LINEs) comprise > 20% of repetitive bases, with a smaller number of DNA transposons, simple repeats, and long terminal repeats (LTRs) identified in each assembly.

## Genome annotation

The *C. forsteri* genome assembly was predicted to comprise a total of 8,564 protein-coding genes (covering 41.93% of the genome) and coding regions (CDSs), which cover 15.29% of the repeat-masked genome (Table 3). The vast majority (95.71%) of CDSs are complete, while the remaining are missing either a start or stop codon, or both. The average predicted gene and CDS length are 10,618 bp and 1,651 bp, respectively, with an average of 6.7 exons per gene. The predicted genes were translated into 8,564 hypothetical polypeptide sequences, 6,620 (77.30%) of which matched to the Swiss-Prot database, with 5,837 unique matches. The hypothetical polypeptide sequences (gene set) and their transcripts comprised 60.27% and 57.75% complete metazoan BUSCOs, respectively (Fig 2B). These results are mostly consistent with those of *Schistosoma* spp. assemblies, except that the average gene, intron, and exon length is smaller [58]. Furthermore, 70 putative proteases, 47 glycosyltransferases, 14 ribonucleases, 10 calpains, 6 cation channels, 6 tetraspanins (TSPs), 2 glutathione *S*-transferases (GSTs), 2 TGF-β homologs, and a fatty-acid binding protein (FABP) were functionally annotated, based on matches to protein sequences in the National Center for Biotechnology Information's (NCBI's) non-redundant protein sequences (NR) database (S2 Table). These protein families were selected as candidate immunogens and drug targets based on research into other digeneans. Matches were primarily to *S. japonicum*, *S. mansoni*, and *S. haematobium*, as well as some to *F. hepatica* and *Clonorchis sinensis*, which fits with known evolutionary relationships, as all are digeneans. Proteins of interest include calpains 1 and 2, secreted cathepsins B and L, and a CD63-like TSP with 4 predicted transmembrane domains (S2 Table).

## Discussion

This is the first draft genome assembly of *C. forsteri* and the first of an aporocotylid. Most of the currently available genomes of trematodes belong to *Schistosoma* and *Fasciola* spp. as etiological agents of neglected tropical diseases (NTDs). *C. forsteri* is a significant pathogen of bluefin tuna and a member of the Schistosomatoidea but differs from its schistosome relatives in that it is monoecious and without suckers [11]. While schistosomes provide the closest point of genomic comparison, the species are expected to be sufficiently divergent to warrant a *de novo* genome assembly. Nanopore long reads were incorporated in this hybrid *de novo*

assembly, which aided in spanning the large repeat regions to greatly improve the contiguity and completeness of the hybrid assembly over short-read assemblies alone.

*In silico* methods for estimating genome size from *k*-mer depth varied greatly in their results, indicating that the *C. forsteri* genome is between 230 and 600 Mb. While *C. forsteri* was assumed to be diploid throughout these analyses based on the Smudgeplot analysis and the adult life cycle stage of schistosomes [59], this remains unknown, and polyploidy might have confounded these estimates. Ploidy varies among platyhelminths, sometimes even within species, so polyploidy would not be unusual for *C. forsteri* [60–62]. Additionally, the large number of repetitive elements typical of trematodes (> 55%), which is far higher than the estimates from GenomeScope (12–15%), may have interfered both with estimates of genome size and the assembly itself. Both genome size and ploidy would be more accurately estimated using laboratory techniques, such as flow cytometry coupled with fluorescence-activated cell sorting (FACS) [63, 64]. As the estimated heterozygosity for both specimens is relatively low (< 1%), this is unlikely to complicate their assemblies, so conventional assembly tools (e.g. ABySS) were used, rather than those developed for highly heterozygous genomes [65]. However, as with estimates of genome size, estimates of heterozygosity may not be accurate. Luo et al. [58] reported relatively high heterozygosity of *S. japonicum* (1.05%).

Although the hybrid contigs are highly contiguous due to Wengan's synthetic scaffolding approach [35], the contigs themselves are not scaffolded. However, improvements to the contiguity of the assembly could not be achieved by scaffolding with the available Nanopore sequence data using LongStitch [66], so future assemblies should incorporate mate-pair sequences, optical mapping, or Hi-C data [67]. The contiguity statistics of these hybrid contigs is comparable to scaffolds of early assemblies of other trematode genomes (S1 Table) [16], and the BUSCO analyses indicates a relatively complete and unfragmented assembly. In particular, the proportion of fragmented BUSCOs is low (< 10% for the predicted genome, transcriptome, and proteome) relative to other trematode assemblies.

*Ab initio* and homology-based gene prediction produced 8,564 protein-coding genes spanning > 40% of the assembled *C. forsteri* genome. As with other trematodes, the ratio of intron-to-exon length is large (> 6:1). While gene length is relatively consistent with other digeneans, the proportion of repeats is generally larger (>55%). Longer repeat regions are being identified in more recent assemblies due to the sequencing of higher molecular weight long reads. This phenomenon was evidenced by the most recent *S. mansoni* assembly, for which Buddenborg et al. [19] found an increase of 11% repetitive bases over the previous assembly. Luo et al. [57] showed that the increase in the proportion of repetitive elements in *Fasciola* spp. largely occurs within intergenic regions, so repetitive elements could likewise be contained within intergenic regions of the *C. forsteri* genome. Nevertheless, average CDS length remains relatively consistent among digeneans (~ 1.5 kb). The predicted mRNA transcripts are not supported by RNA-seq data, but the vast majority (> 77%) of polypeptide sequences generated from the predicted gene set match to other trematodes, which fits into an evolutionary context.

As with schistosomiases in humans, the drug of choice for treating bluefin tuna infected with *Cardicola* spp. is PZQ, which is thought to induce paralysis of the parasites by interfering with voltage-gated $Ca^{2+}$ channels (VGCCs) in adult platyhelminths [68, 69]. In this study, 6 putative cation channels were functionally annotated from this *C. forsteri* genome assembly (S2 Table). Park et al. [70] showed that PZQ activated a schistosomal transient receptor potential melastatin ion channel (TRPM, A0A5K4F0X5) *in vitro*, but a single missense mutation (Asn[1388]→Thr) in the hydrophobic binding pocket confers resistance to PZQ, which occurs in *F. hepatica*. Interestingly, the first 54 residues (including Asn[1388]) of the binding pocket do not align with the most homologous putative CfTRPM1 (Table 4, sequence identity = 84.54%), but

**Table 4. Select *Cardicola forsteri* protein homologs and their predicted conserved structural domains.**

| Putative protein | Abbreviation | Pfam-A |
|---|---|---|
| Calpain 1 | CfCalp1 | PF00648, PF01067 |
| Calpain 2 | CfCalp2 | PF00648, PF01067 |
| Cathepsin B | CfCB | PF00112, PF08127 |
| Cathepsin L | CfCL | PF00112, PF08246 |
| 25-kDa GST | Cf25GST | PF14497, PF02798 |
| 27-kDa GST | Cf27GST | PF13417, PF14497 |
| TRPM | CfTRPM1 | PF18139, PF00293, PF00520 |
| TSP-2 (CD63-like) | Cf-TSP-2 | PF00335 |

GST, glutathione *S*-transferase; TRPM, transient receptor potential melastatin; TSP, tetraspanin.

all the remaining residues identified as essential to PZQ-sensitivity are conserved. Therefore, we conclude either that PZQ interacts differently with *C. forsteri* TRPM or that PZQ kills *C. forsteri* via alternative mechanisms. This may indicate that the PZQ mechanism of action is more complex than interactions with a single target.

Most of the putative *C. forsteri* proteins are predicted to be N-glycosylated. Correspondingly, 47 putative glycosyltransferases were identified in this study (S2 Table). As expected, many of these are likely responsible for the synthesis of N-glycans, as indicated by structural domains in the CAZy database [52]. In its adult life cycle stage, *C. forsteri* predominately synthesizes oligomannose N-glycans, as well as paucimannosidic and complex-type N-glycans carrying core fucose and xylose [71]. No α1–3-linked fucoses were found in adults, but 3 putative α1–3 fucosyltransferases were identified in this study (Table 5). In particular, CfFucTD is homologous (sequence identity > 46%) to schistosomal α1–3 fucosyltransferases D (E2EAI7), E (E2EAI8), and F (E2EAI9), which have been functionally characterised and were shown to synthesize Lewis X and fucosylated LacdiNAc motifs [72, 73]. Therefore, *C. forsteri* likely synthesizes these motifs in pre-adult life cycle stages, as postulated by Coff et al. [71]. Sustained IgG1 and IgG3 responses to cercarial multifucosylated LDN motifs have been associated with protective immunity against schistosomiases [74–77], so glycomics is crucial to a holistic

**Table 5. Select putative *Cardicola forsteri* glycosyltransferases with their predicted structural domains.**

| Putative protein | Abbreviation | Pfam-A | CAZy |
|---|---|---|---|
| **Fucosyltransferases** | | | |
| Peptide–O-fucosyltransferase | CfFucTA | PF10250 | – |
| Galactoside α1–3 fucosyltransferase | CfFucTB | PF00852, PF17039 | GT10 |
| | CfFucTC | PF00852, PF17039 | GT10 |
| | CfFucTD | PF00852, PF17039 | GT10 |
| α1–6-fucosyltransferase | CfFucTE | PF19745 | NA |
| **Xylosyltransferases** | | | |
| Xylosyltransferase I | CfXylT1 | PF02485, PF12529 | GT13 |
| Glycoprotein β1–2 xylosyltransferase | CfXylT2 | PF04577 | GT61 |
| ***N*-acetylgalactosyltransferases** | | | |
| β1–4 *N*-acetylgalactosaminyltransferase | CfGalNT1 | PF13733, PF02709 | GT7 |
| **Glucuronyltransferases** | | | |
| Bifunctional β1–3 glucuronyltransferase | CfGlcAT1 | PF03360 | GT43 |
| | CfGlcAT2 | PF13896 | GT49 |

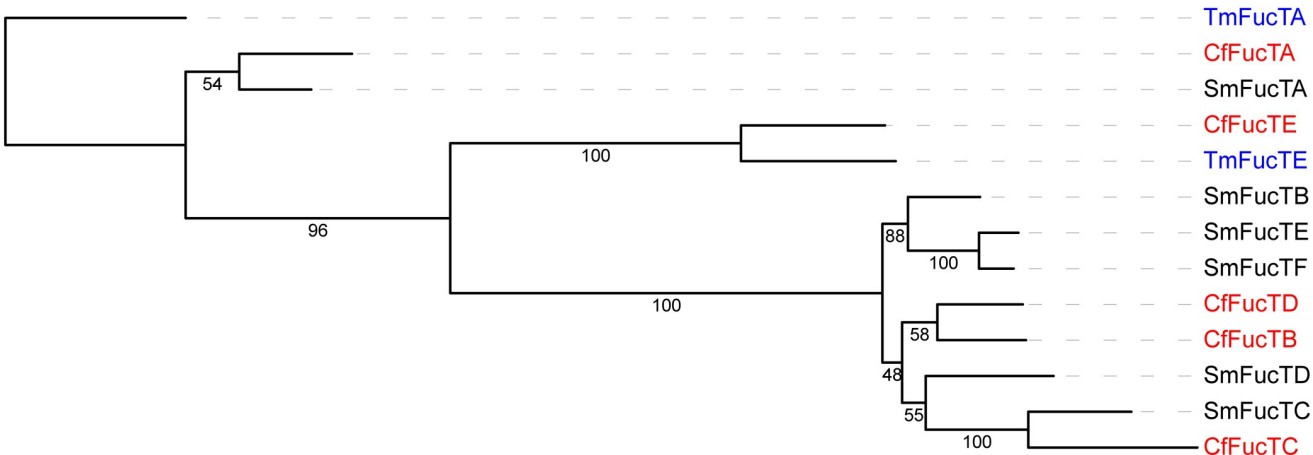

**Fig 3. Phylogenetic tree of *Cardicola forsteri* fucosyltransferases.** Outgroup-rooted maximum likelihood tree of the fucosyltransferases from *Cardicola forsteri* (shown in red) in Table 5 with fucosyltransferases from *Schistosoma mansoni* and *Thunnus maccoyii* (as the outgroup in blue): SmFucTA (G9HW08), SmFucTB (E2EAI5), SmFucTC (E2EAI6), SmFucTD (E2EAI7), SmFucTE (E2EAI8), SmFucTF (E2EAI9), TmFucTA (XP_042266362.1), TmFucTE (XP_042246885.1). Bootstrap values are shown on the branches.

understanding of host–parasite interactions. Interestingly, CfFucTE is grouped with a hypothetical α1–6 fucosyltransferase from SBT (XP_042246885.1) in the phylogenetic analysis (Fig 3), although this node shares only 34% sequence identity. Additionally, several putative O-linked glycosyltransferases were identified, indicating that, like its schistosome relatives, *C. forsteri* may also synthesize O-glycans in its pre-adult life cycle stages [78, 79]. For example, CfFucTA shares homology (sequence identity = 51.51%) with *S. mansoni* protein–O-fucosyltransferase A (G9HW08), although these two proteins are grouped with low bootstrap support (54%). Two putative glucuronyltransferases were also identified, which could be involved in the synthesis of hexuronic acid-carrying N-glycans found in adult *C. forsteri*. However, glucuronyltransferases are not well characterised, so these require further functional studies. Unusually for a trematode, adult *C. forsteri* synthesizes core-xylose carrying N-glycans, and a putative β1–2 xylosyltransferase (CfXylT2) was also identified. Both β1–2-linked xyloses and glycosyltransferases elicit a humoral immune response in mammals following infection with *S. mansoni* [77, 80].

Cathepsins are a family of proteases that are important virulence factors in trematodes, facilitating tissues invasion, digestion, and immune evasion [81]. Cathepsins B and L (cysteine proteases) are secreted in high concentrations and often elicit humoral immune responses. These cathepsins, particularly *F. hepatica* cathepsin L1 (Q7JNQ9), have been the target of vaccination of ruminants against fascioliases [82]. A putative *C. forsteri* protease was classified as CfCL (Table 4) from sequence identity (44.68%) and a conserved active site (Cys$^{134}$, His$^{281}$, and Asn$^{301}$), which clusters with cathepsin L1 of schistosomes (Fig 4A). Buffoni et al. [83] postulated that 42 residues within FhCL1 are responsible for protective immunity in vaccinated sheep, however these are not well conserved in CfCL, which contains substantial non-conservative substitutions. The prime schistosomal cathepsin targeted in vaccine studies is the Sm31 antigen *S. mansoni* cathepsin B1 (P25792), which is an abundant digestive enzyme secreted into the gut [84–88]. However, CfCB shares closest homology (sequence identity = 69.65%) and a phylogenetic node (Fig 4B) with SmCB2 (Q95PM1) [89], which was reactive to IgG from mice both infected or vaccinated with *S. mansoni* cercariae [80]. Additionally, *S. japonicum* cathepsin B2 (Q7Z1I6) is thought to be involved in skin penetration by the cercariae and degradation of host immune proteins [90]. As essential secreted gut antigens, cathepsins are

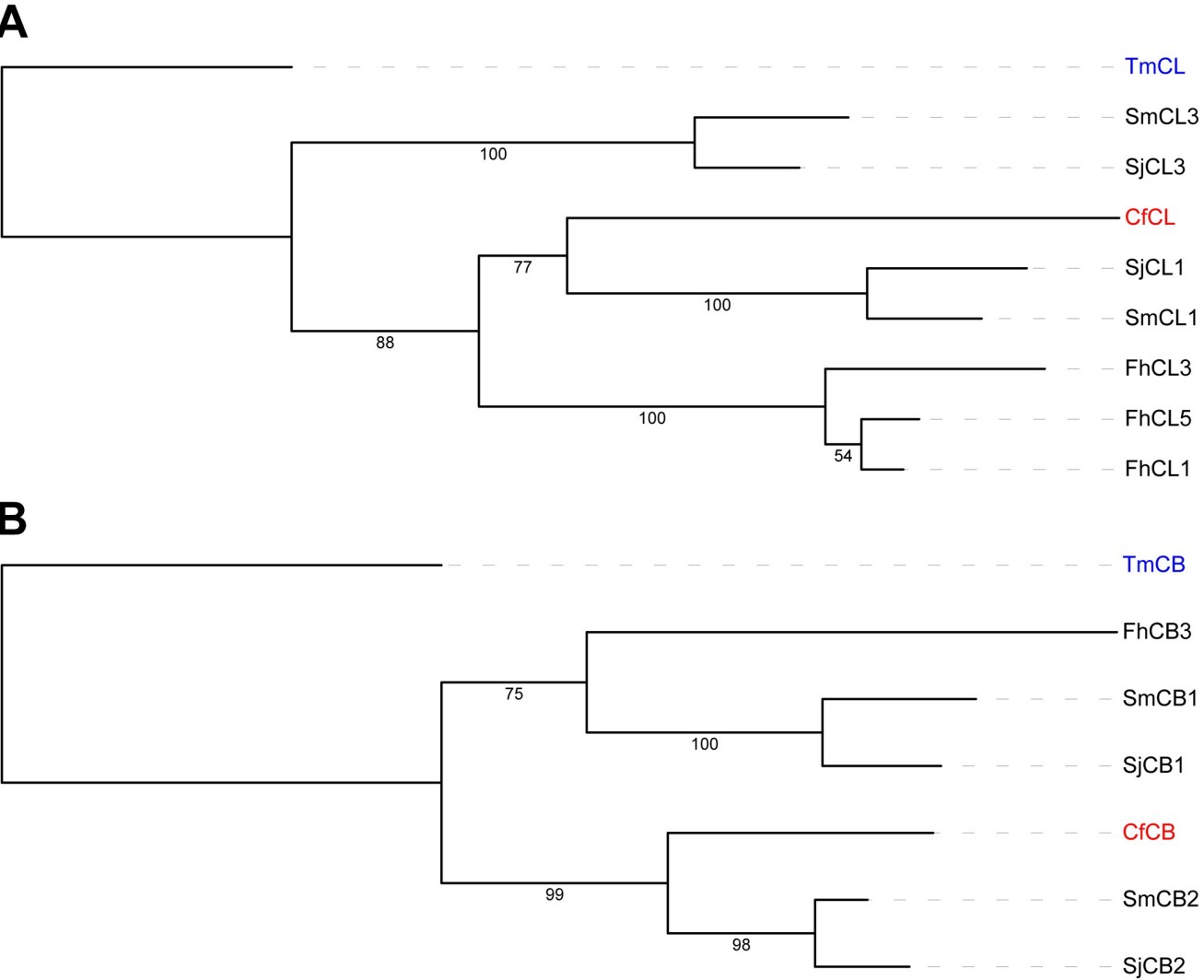

**Fig 4. Phylogenetic tree of *Cardicola forsteri* cathepsins.** Outgroup-rooted maximum likelihood tree of the cathepsins from *Cardicola forsteri* (shown in red) in Table 4 with cathepsins from *Schistosoma mansoni*, *Schistosoma japonicum*, *Fasciola hepatica*, and *Thunnus maccoyii* (as the outgroup in blue). Bootstrap values are shown on the branches. (A) Cathepsins L, including: SmCL1 (Q26564), SmCL3 (B4XC67), SjCL1 (C1LBR4), SjCL3 (C1LJC3), FhCL1 (Q7JNQ9), FhCL3 (B9VXS1), FhCL5 (A0A2H4PKA2), TmCL (XP_042278099.1). (B) Cathepsins B, including: SmCB1 (P25792), SmCB2 (Q95PM1), SjCB1 (P43157), SjCB2 (Q7Z1I6), FhB3 (A7UNB2), TmCB (XP_042261887.1).

promising drug and vaccine targets. However, Farias et al. [80] found few epitopes within the active centre of schistosomal cathepsins, so direct inhibition is likely limited, which could explain the relative lack of success with cathepsin-based vaccines against schistosomiases compared with those against fascioliases. While immunogenicity is reportedly highest against gut antigens, tegumental proteins are also at the host–parasite interface and have been the subject of helminth vaccinology for decades.

Highly immunoreactive, calpains are $Ca^{2+}$-dependent cysteine proteases that participate in a variety of biological functions. Wang et al. [91] showed that two calpains, SmCalp1 (P27730.1) and SmCalp2 (ATN96084.1), are expressed on the tegument of adult *S. mansoni* and schistosomula, where they cleave host fibronectin, thereby preventing blood clotting. Two putative *C. forsteri* calpains were classified as CfCalp1 and CfCalp2 (Table 4), based on conserved structural domains and sequence identity (54.19% and 79.94%, respectively). Among

platyhelminths, amino acid sequences of calpains diverge greatly, however a Cys, His, and Asn residue are conserved in the active site of calpain 1. The His residue is replaced with Gln in calpain 2, and all these residues are conserved in CfCalp1 (Cys[151], His[310], and Asn[334]) and CfCalp2 (Cys[116], Gln[282], and Asn[310]). Since 1997, SmCalp1 (the Smp80 antigen) has been a target of vaccination against schistosomiasis [92], and promising results from pre-clinical trials have been published more recently [93, 94]. Other putative tegumental proteins of *C. forsteri* that could be viable vaccine candidates include GSTs, TSPs, and an FABP.

Similar to *S. mansoni* and *S. japonicum*, putative 25-kDa and 27-kDa GSTs were identified from the *C. forsteri* genome (Table 4), with homology to Class μ and Class ω GSTs, respectively [95]. As crucial detoxification enzymes, GSTs are found on the tegument of schistosomes and are frequently targeted by novel vaccines and drugs. The most advanced of these is Sh28GST (P30114), which is the target of the Bilhvax vaccine against urinary schistosomiasis [96, 97]. However, research into rSh28GST has stalled following Phase III trial results that reported a lack of efficacy in children [98]. While neither of the two putative *C. forsteri* GSTs shares close homology with Sh28GST (sequence identity < 27%), Tyr[7] and Arg[18] are conserved in Cf25GST, and Tyr[10] and Arg[21] of Sh28GST were identified by Angelucci et al. [99] to form π-cation interactions with one another. Another tegumental target proceeding through vaccine trials is SmTSP-2 (Q8ITD7), a CD63-like tetraspanin that is thought to be involved in tegument formation and extracellular vesicle secretion [100, 101]. Of the 6 putative *C. forsteri* tetraspanins (S2 Table), Cf-TSP-2 was predicted to be CD63-like, with low homology to SmTSP-2 (sequence identity = 33.18%). Additionally, a putative 15-kDa FABP was identified, with homology (sequence identity = 56.49%) to the Sm14 FABP antigen (P29498), which is currently the target of Phase II clinical trials [102, 103].

Protective immunity to helminthiases is inhibited by immunomodulatory mechanisms deployed by the parasite, which polarise an anti-inflammatory $T_H2$ response and induce T cell anergy. These host–parasite interactions are not fully understood but involve parasite glyco-conjugates interacting with host C-type lectin receptors (CLRs) and toll-like receptors (TLRs) [104]. The best described of these interactions is of schistosomal soluble egg antigens (SEA), namely the glycosylated T2 ribonuclease ω-1, interacting with mannose receptors (MRs) on monocyte-derived dendritic cells (Mo-DCs) [105]. No homologs of ω-1 were found in the *C. forsteri* genome, but two proteins contain a TGF-β-like domain (S2 Table), and TGF-β homologs are known to be expressed by other helminths to modulate host immunity [106, 107]. Nevertheless, potential mechanisms of evasion and modulation of the bluefin tuna immune responses by *C. forsteri* require further investigation.

The whole genome of *C. forsteri* was sequenced and assembled for the first time, into high-quality contigs. The highly repetitive genome was functionally annotated with putative glycosyltransferases and some potential vaccine targets, including calpains and an FABP. Future research should be directed toward functionally characterising these putative proteins using *in vitro* assays to further understand this parasite and its interactions with bluefin tuna. Additionally, this genome will provide a framework for genomic investigations into other *Cardicola* spp. to advance our understanding of their susceptibility to PZQ and the development of alternative control measures.

## Supporting information

**S1 Table. *Cardicola forsteri* assembly statistics.** Comparison between the assembly statistics of *Cardicola forsteri* and related digeneans. Scaffold statistics are not applicable to the *C. forsteri* assembly, which is contig-level. Benchmarking Universal Single-Copy Orthologs (BUS-COs) were assessed using BUSCO v5.2.2 [37] against the metazoa_odb10 database. Both the

first and most recent assemblies of the related digeneans are included.
(XLSX)

**S2 Table. Functional annotation of the *Cardicola forsteri* gene set.** Functional annotation of select *Cardicola forsteri* genes. The closest matches to the National Center for Biotechnology Information's (NCBI's) non-redundant protein sequences (NR), Pfam v35.0 [51], and CAZy [52] databases are included. Predicted N-glycosylations sites, transmembrane domains, and secretion modes are also included, where relevant. Abbreviations are given, where relevant to genes discussed in the manuscript.
(XLSX)

# Acknowledgments

We thank the Australian Southern Bluefin Tuna Industry Association and commercial tuna companies for their assistance and support in sample collection. This research was supported by The University of Melbourne's Research Computing Services and the Petascale Campus Initiative.

# Author Contributions

**Conceptualization:** Lachlan Coff, Barbara F. Nowak, Paul A. Ramsland, Nathan J. Bott.

**Data curation:** Lachlan Coff.

**Formal analysis:** Lachlan Coff, Andrew J. Guy, Bronwyn E. Campbell.

**Funding acquisition:** Nathan J. Bott.

**Investigation:** Lachlan Coff.

**Methodology:** Lachlan Coff, Andrew J. Guy, Bronwyn E. Campbell, Barbara F. Nowak, Paul A. Ramsland.

**Project administration:** Nathan J. Bott.

**Supervision:** Paul A. Ramsland, Nathan J. Bott.

**Validation:** Paul A. Ramsland.

**Visualization:** Barbara F. Nowak, Paul A. Ramsland, Nathan J. Bott.

**Writing – original draft:** Lachlan Coff.

**Writing – review & editing:** Andrew J. Guy, Bronwyn E. Campbell, Barbara F. Nowak, Paul A. Ramsland, Nathan J. Bott.

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
