## [Decision Letter · Decision Letter 0]

19 Sep 2022

PONE-D-22-18941Draft genome of the bluefin tuna blood fluke, CardicolaforsteriPLOS ONE

Dear Dr. Bott,

Thank you for submitting your manuscript to PLOS ONE. After careful consideration, we feel that it has merit but does not fully meet PLOS ONE’s publication criteria as it currently stands. Therefore, we invite you to submit a revised version of the manuscript that addresses the points raised during the review process.

We look forward to receiving your revised manuscript.

Kind regards,

Hudson Alves Pinto, Ph.D

Academic Editor

PLOS ONE

Journal Requirements:

Additional Editor Comments:

I congratulate the authors for this interesting and relevant contribution for field of ichtyoparasitology.

Reviewers' comments:

Reviewer's Responses to Questions

**Comments to the Author**

1. Is the manuscript technically sound, and do the data support the conclusions?

Reviewer #1: Yes

Reviewer #2: Yes

2. Has the statistical analysis been performed appropriately and rigorously? 

Reviewer #1: Yes

Reviewer #2: Yes

3. Have the authors made all data underlying the findings in their manuscript fully available?

Reviewer #1: Yes

Reviewer #2: Yes

4. Is the manuscript presented in an intelligible fashion and written in standard English?

Reviewer #1: Yes

Reviewer #2: Yes

5. Review Comments to the Author

Reviewer #1: The article by Coff et al. presents a high-quality genome of Cardicola fosteri, a much needed genomic resource for an important parasite of aquacultured fish. The authors highlight the value of this resource through their identification and characterisation of known proteins associated with host:parasite interactions or that have been identified as promising drug targets or vaccine candidates. The only limitation of this study was a lack of transcriptomic evidence of species-specific genes that may have reduced the number of overall genes predicted from the genome. Overall, I thought he article was excellent and aligns with the scope of this journal.

I had some minor comments or suggestions for consideration below.

Lines 29 to 32. I suggest a simplified outline of co-infecting Cardicola spp. and their importance in the introduction before focussing on fosteri.

Lines 45 to 54 could be further summarised. Or removed. FOcus on Cardicola and importance of a genomic resource is fine.

seven autosomes? and the sex chromosomes?

Lines 62 to 66. Suggest better integration of knowledge gaps that could be addressed with genomic data. The glycome section seems out of place in an introduction (but great for discussion).

Abbreaviate species names upon first use throughout

Lines 86 to 89. Fine to just state that you had ethics approval without stating that it was not required.

Line 168. given the wealth of proteomic data for Platyhelminthes, why was ProtHint only provided with S. mansoni? This is of particular importance considering the lack of transcriptomic

data for C. fosteri

Line 187, Please specify the substitution model used.

Line 192. WIll the data for specimen 1 be made public? Raw data and/or assembly?

Line 221. It is okay to refer to the "hybrid contigs" as scaffolds as long as the DNA was from the same species. Hybrid makes it sound like you used a different species. Perhaps add a clear definition of hybrid?

Line 225.A comparison of completeness when compared to other flatworms is important. No flatworm will have 100% as many genes appear to be absent or lack sequence homology for BUSCO

Detection compared to other metazoans. I think you are underselling the completeness of your genome here.

Line 280. It would be possible to use PloipY or SmudgePlot with your GenomeScope results to estimate ploidy. As this is an

important discussion point, it is worth considering this additional

analysis.

Lack of transcriptomic data limits the conclusions from the proteins

characterised herein (e.g. important to know in which stage of the parasite they are being expressed)

Lines 423 to 427. Would have been nice to hear something about the other Cardicola species and the future directions in the concluding

Reviewer #2: I consider that the manuscript is suitable for publication in this journal, mainly because it is complete, very well written, and because of the importance it provides for parasitology. This type of work is a significant advance in the parasitology applied to fish, mainly due to the impact that the species of the parasite under study can have on the economy of the fishing area. I believe that its justification and the possible application of the results obtained for the management and control of Cardicola fosteri in the future are very convenient.

6. PLOS authors have the option to publish the peer review history of their article (what does this mean?). If published, this will include your full peer review and any attached files.

Reviewer #1: No

Reviewer #2: No

---

## [Author Response · Author response to Decision Letter 0]

27 Sep 2022

Reviewer #1: The article by Coff et al. presents a high-quality genome of Cardicola fosteri, a much needed genomic resource for an important parasite of aquacultured fish. The authors highlight the value of this resource through their identification and characterisation of known proteins associated with host:parasite interactions or that have been identified as promising drug targets or vaccine candidates. The only limitation of this study was a lack of transcriptomic evidence of species-specific genes that may have reduced the number of overall genes predicted from the genome. Overall, I thought he article was excellent and aligns with the scope of this journal.

I had some minor comments or suggestions for consideration below.

Lines 29 to 32. I suggest a simplified outline of co-infecting Cardicola spp. and their importance in the introduction before focussing on fosteri.

Response: We believe this outline is important for establishing the distinctions between the co-infecting Cardicola spp., Thunnus spp., and blood fluke infections of bluefin tuna in different parts of the world. The focus on C. forsteri is justified by it being the dominant species infecting bluefin tuna.

Lines 45 to 54 could be further summarised. Or removed. FOcus on Cardicola and importance of a genomic resource is fine. seven autosomes? and the sex chromosomes?

Response: ‘Seven chromosomes’ has been corrected to ‘seven autosomes and two sex chromosomes’ on p. 4, ln. 56–57 of the revised manuscript, in accordance with the reviewer’s comment.

Lines 62 to 66. Suggest better integration of knowledge gaps that could be addressed with genomic data. The glycome section seems out of place in an introduction (but great for discussion).

Response: Glycomics has been removed from this section of the Introduction, in accordance with the reviewer’s comment.

Abbreaviate species names upon first use throughout

Response: Species names have been abbreviated after their first appearance throughout the manuscript, with the exception of headings and legends, in accordance with the reviewer’s comment.

Lines 86 to 89. Fine to just state that you had ethics approval without stating that it was not required.

Response: The ethics statement has been amended on p. 5, ln. 84–86 of the revised manuscript, in accordance with the reviewer’s comment.

Line 168. given the wealth of proteomic data for Platyhelminthes, why was ProtHint only provided with S. mansoni? This is of particular importance considering the lack of transcriptomic data for C. fosteri

Response: The S. mansoni proteome was selected as the protein database for ProtHint because it is the closest relative of C. forsteri with a published genome annotation, which has been carefully curated for over ten years. Other proteomes from platyhelminths could have been included as a reference, but we decided against this for two reasons. The first was to reduce the size of this protein database to optimize computational efficiency. The second was to reduce bias in the annotation from lower-quality reference annotations. Brůna et al. 2021 showed that the accuracy of their BRAKER2 gene prediction pipeline improved when ProtHint was provided with proteomes of closer evolutionary distance. They refer to this approach as ‘BRAKER2 with proteins of short evolutionary distance’, which is what we employed for this manuscript. They also showed that a larger database of proteins could compensate for evolutionary distance, but only outside of the Family. It is therefore not expected that the accuracy of gene prediction would substantially increase with the addition of reference proteins from more evolutionarily distant platyhelminths, although this could be tested for future assemblies.

Line 187, Please specify the substitution model used.

Response: The substitution model has been specified as ‘the WAG amino acid substitution model’ on p. 9, ln. 189–190 of the revised manuscript, in accordance with the reviewer’s comment.

Line 192. WIll the data for specimen 1 be made public? Raw data and/or assembly?

Response: The raw sequence data for Specimen 1 will be made public upon publication of this manuscript, having been submitted to the NCBI’s Sequence Read Archive.

Line 221. It is okay to refer to the "hybrid contigs" as scaffolds as long as the DNA was from the same species. Hybrid makes it sound like you used a different species. Perhaps add a clear definition of hybrid?

Response: ‘Hybrid’ has been defined as ‘assembled from short and long reads’ as per Di Genova et al. 2021 on page 11, ln. 225–226 of the revised manuscript, in accordance with the reviewer’s comment.

Line 225.A comparison of completeness when compared to other flatworms is important. No flatworm will have 100% as many genes appear to be absent or lack sequence homology for BUSCO Detection compared to other metazoans. I think you are underselling the completeness of your genome here.

Response: Comparisons to the BUSCO analyses of Fasciola hepatica and Schistosoma mansoni have been added to p. 11, ln. 230–232 of the revised manuscript, in accordance with the reviewer’s comment.

Line 280. It would be possible to use PloipY or SmudgePlot with your GenomeScope results to estimate ploidy. As this is an important discussion point, it is worth considering this additional analysis.

Response: A Smudgeplot analysis was conducted, which proposed C. forsteri to be diploid in its adult life cycle stage. This result was included in the revised manuscript, in accordance with the reviewer’s comment.

Lack of transcriptomic data limits the conclusions from the proteins characterised herein (e.g. important to know in which stage of the parasite they are being expressed)

Response: This is a point for further investigation discussed on p. 15, ln 315–317 of the revised manuscript. This study was limited by collection of sufficient quantities of the parasite from the field, and it would be difficult to collect different stages of the parasite in such a manner. Therefore, a study of this kind would need to be funded and carefully planned in coordination with industry beforehand.

Lines 423 to 427. Would have been nice to hear something about the other Cardicola species and the future directions in the concluding

Response: A brief statement outlining the impact of this study for other Cardicola spp. has been added to the concluding remarks of the Discussion on p. 20, ln. 432–434 of the revised manuscript, in accordance with the reviewer’s comment.

Reviewer #2: I consider that the manuscript is suitable for publication in this journal, mainly because it is complete, very well written, and because of the importance it provides for parasitology. This type of work is a significant advance in the parasitology applied to fish, mainly due to the impact that the species of the parasite under study can have on the economy of the fishing area. I believe that its justification and the possible application of the results obtained for the management and control of Cardicola fosteri in the future are very convenient.

---

## [Editor Report · Decision Letter 1]

4 Oct 2022

Draft genome of the bluefin tuna blood fluke, Cardicola forsteri

PONE-D-22-18941R1

Dear Dr. Bott,

We’re pleased to inform you that your manuscript has been judged scientifically suitable for publication and will be formally accepted for publication once it meets all outstanding technical requirements.

Kind regards,

Hudson Alves Pinto, Ph.D

Academic Editor

PLOS ONE
---

## [Editor Report · Acceptance letter]

6 Oct 2022

PONE-D-22-18941R1 

Draft genome of the bluefin tuna blood fluke, *Cardicola forsteri*

Dear Dr. Bott:

I'm pleased to inform you that your manuscript has been deemed suitable for publication in PLOS ONE. Congratulations! Your manuscript is now with our production department. 

Kind regards, 

on behalf of

Dr. Hudson Alves Pinto 

Academic Editor

PLOS ONE